

**Implementation of state-of-the-art ternary new particle formation scheme to**
**the regional chemical transport model PMCAMx-UF in Europe**
Elham Baranizadeh[1], Benjamin N. Murphy[2,3], Jan Julin[1,2], Saeed Falahat[2,4], Carly L. Reddington[5], Antti
Arola[6], Santtu Mikkonen[1], Christos Fountoukis[7], David Patoulias[8], Andreas Minikin[9], Thomas
Hamburger[10], Ari Laaksonen[1,11], Spyros N. Pandis[8,12,13], Hanna Vehkamäki[14], Kari E. J. Lehtinen[1,6],
Ilona Riipinen[2]
[1]Department of Applied Physics, University of Eastern Finland, Kuopio, Finland
[2]Department of Environmental Science and Analytical Chemistry (ACES), Stockholm University,
Stockholm, Sweden
[3]Now at the National Exposure Research Laboratory, US Environmental Protection Agency, Research
Triangle Park, USA
[4]Now at the Swedish Meteorological and Hydrological institute (SMHI), Norrköping, Sweden
[5]Institute for Climate and Atmospheric Science, School of Earth and Environment, University of Leeds,
Leeds, UK
[6]Finnish Meteorological Institute, Kuopio, Finland
[7]Qatar Environment and Energy Research Institute (QEERI), Hamad Bin Khalifa University (HBKU),
Qatar Foundation, Doha, Qatar
[8]Department of Chemical Engineering, University of Patras, Patras, Greece
[9]German Aerospace Agency DLR, Oberpfaffenhofen, Germany
[10]Norwegian Institute for Air Research (NILU), Oslo, Norway
[11]Climate research Unit, Finnish Meteorological Institute, Helsinki, Finland
[12]Institute of Chemical Engineering Sciences, Foundation for Research and Technology Hellas
(ICEHT/FORTH), Patras, Greece
[13]Department of Chemical Engineering, Carnegie Mellon University, Pittsburgh, PA, USA
[14]Division of Atmospheric Sciences, Department of Physics, University of Helsinki, Helsinki, Finland

**Abstract**
The particle formation scheme within PMCAMx-UF, a three dimensional chemical transport model,
was updated with particle formation rates for the ternary $H_2SO_4$-$NH_3$-$H_2O$ pathway simulated by the
Atmospheric Cluster Dynamics Code (ACDC) using quantum chemical input data. The model was
applied over Europe for May 2008, during which the EUCAARI-LONGREX campaign was carried out
providing aircraft vertical profiles of aerosol number concentrations. The updated model reproduces the
observed number concentrations of particles larger than 4 nm within one order of magnitude throughout
the atmospheric column. This reasonable agreement is very encouraging considering the fact that no





semi-empirical fitting was needed to obtain realistic particle formation rates. The cloud adjustment
scheme for modifying the photolysis rate profiles within PMCAMx-UF was also updated with the TUV
(Tropospheric Ultraviolet and Visible) radiative transfer model. Results show that although the effect of
the new cloud adjustment scheme on total number concentrations is small, enhanced new particle
formation is predicted near cloudy regions. This is due to the enhanced radiation above and in the
vicinity of the clouds, which in turn leads to higher production of sulfuric acid. The sensitivity of the
results to including emissions from natural sources is also discussed.

## 45  1 Introduction

Formation of new particles from atmospheric vapors (new particle formation, NPF) is potentially an
important source of particulate matter in the atmosphere, especially in the ultrafine (<100 nm in
diameter) size range (Kulmala et al., 2004; Merikanto et al., 2009; Jung et al., 2010; Fountoukis et al.,
2012; Kerminen et al., 2012; Fuzzi et al, 2015). In the past, in modeling studies on the role of in-situ
NPF as a particle source, particle formation has been represented with various parameterizations
including binary (Vehkamäki et al., 2002) or ternary (Napari et al., 2002) nucleation based on the
classical nucleation theory (CNT), semi-empirical activation (Kulmala et al., 2006), kinetic (McMurry,
1980) or organic-enhanced (Paasonen et al., 2010) NPF and/or ion mediated nucleation (Yu and Luo,
2009). These parameterizations have generally assumed sulfuric acid ($H_2SO_4$), water ($H_2O$), ammonia
($NH_3$), or different organic species as the compounds forming the new particles. The activation, kinetic
and organic-enhanced mechanisms are semi-empirical, based on the observed dependence of particle
formation rates on concentrations of sulfuric acid and/or organic vapors (Sihto et al., 2006; Paasonen et
al., 2010). The advantage of such methods is that they are simple and produce nucleation rates of the
same order as those observed. However, as they are fit to specific experiments usually at ground level,
they are most reliable at locations and conditions similar to those at which the data has been obtained.
The ternary $H_2SO_4$-$H_2O$-$NH_3$ parameterization by Napari et al. (2002) has been used with some success
(Adams and Seinfeld, 2002; Jung et al., 2008; Jung et al., 2010; Fountoukis et al., 2012; Westervelt et
al., 2014), but with quite drastic correction factors necessary to reproduce ambient particle number
concentrations. In many previous studies (Spracklen et al., 2006; Makkonen et al., 2009; Yu et al., 2010)
the binary $H_2SO_4$-$H_2O$ nucleation has been assumed to dominate in the upper atmosphere and be
negligible at lower altitudes, and it has often been superimposed with one of the other mechanisms.

Sulfuric acid, water and ammonia have long been established as important molecules forming new
particles in the atmosphere (Korhonen et al., 1999; Kulmala et al., 2000; Laaksonen et al., 2008).
However, standard theoretical descriptions of the ternary $H_2SO_4$-$H_2O$-$NH_3$ particle formation pathway
have not been able to reproduce measured particle formation rates – hence the need to resort to semi-
empirical parameterizations and correction factors to describe this process in atmospheric models.



Recent experimental (Kirkby et al., 2011; Almeida et al., 2013; Jen et al., 2014) and computational
developments have, however, changed this picture drastically. Flexible computational models (such as
the Atmospheric Cluster Dynamics Code, ACDC, Olenius et al., 2013) which simulate the kinetics of a
population of molecular clusters combined with cluster free energies calculated from first-principles
methods, can now reproduce laboratory observations of particle formation rates in $H_2SO_4$-$NH_3$ as well
as $H_2SO_4$-amine systems with reasonable accuracy (Almeida et al., 2013), without the need for empirical
scaling of the predicted particle formation rate.

Predictions of particle number concentration from regional-scale chemical transport models have been
evaluated typically with data from ground-level observations (Jung et al., 2008; Matsui et al., 2011a,
2013c; Fountoukis et al., 2012; Cui et al., 2014; Lupascu et al., 2015). Meanwhile, there is much to gain
from assessing the model against vertically-resolved particle number observations, as many of the
uncertainties in the model relate to particle scavenging, by hydrometeors as well as other particles, and
mixing of air masses. The possible biases introduced from parameterizing new particle formation rates
with ground-level data makes it all the more imperative to evaluate and constrain models with
observations taken at altitude. Recent studies (Reddington et al., 2011; Lupascu et al., 2015) have begun
assessing global- and regional-scale models in this way against data from European and US field
campaigns involving aircraft measurements. Furthermore, it is worthwhile to explore the vertical
variability in chemical and environmental precursors to NPF (e.g. $H_2SO_4$, $NH_3$, $T$, RH, etc.) and particle
number concentrations.

In this work we describe the implementation of a $H_2SO_4$-$H_2O$-$NH_3$ new particle formation scheme based
on the output of the ACDC model to the regional chemical transport model PMCAMx-UF (Jung et al.,
2010, Fountoukis et al., 2012). We test the new scheme by simulating the evolution of atmospheric gas-
phase and aerosol particle concentrations during May 2008 in Europe. We evaluate the model against
ground-based and airborne observations of aerosol particle number size distributions during the
simulated period. Furthermore, we implement an updated radiative transfer scheme TUV (Tropospheric
Ultraviolet and Visible radiative transfer model; Madronich, 2002) for PMCAMx-UF and discuss its
implications for predictions of NPF and particle number concentrations in the European domain.

**2 Methods**
**2.1 PMCAMx-UF model description**
PMCAMx-UF is a three-dimensional regional chemical transport model that simulates both the size-
dependent particle number and chemically-resolved mass concentrations (Jung et al. 2010). PMCAMx-
UF utilizes the framework of the air quality model PMCAMx (Gaydos et al., 2007, Karydis et al., 2007),



where the description of vertical and horizontal advection and dispersion, wet and dry deposition, and
gas-phase chemistry are based on the CAMx air quality model, and the variable size-resolution model
of Fahey and Pandis (2001) is used for aqueous-phase chemistry. To treat the aerosol microphysics,
including NPF, condensation and coagulation, PMCAMx-UF uses the Dynamic Model for Aerosol
Nucleation (DMAN) module by Jung et al. (2006). DMAN uses the Two-Moment Aerosol Sectional
(TOMAS) algorithm (Adams and Seinfeld, 2002) to track the aerosol number and mass distributions.
DMAN divides the aerosol particles into 41 logarithmically-spaced size bins between 0.8 nm and 10
μm.

The aerosol species modeled in PMCAMx-UF include sulfate, ammonium, water, elemental carbon,
crustal material, chloride, sodium, nitrate, primary organic aerosol and four secondary organic aerosol
surrogate compounds. The version of TOMAS used in the model applied here tracks explicitly the mass
transfer of sulfate and ammonium while that of water is treated assuming equilibrium. Within the
DMAN aerosol microphysics module the remaining compounds are represented by inert surrogate
species. The pseudo-steady-state approximation method (Pierce and Adams, 2009), which assumes
steady-state concentration for sulfuric acid, is used for the calculation of NPF and sulfuric acid
condensation rates. The condensation of ammonia is calculated independently following the approach
described in Jung et al. (2006).

New particle formation rates in the standard version of PMCAMx-UF have been calculated in previous
studies using a scaled version of the ternary $H_2SO_4$-$NH_3$-$H_2O$ parametrization by Napari et al. (2002),
hereafter referred to as the "scaled" Napari parameterization. The original Napari parameterization is
based on predictions of the CNT assuming that the energetics of the molecular clusters follow bulk
thermodynamics. While it has been shown to perform better than a range of other nucleation
parameterizations in predicting the occurrence of new particle formation events (Jung et al. 2008), it is
also known to overpredict ultrafine particle number concentrations (Gaydos et al., 2005; Yu et al.,
2006a; Jung et al., 2006; Merikanto et al., 2007b; Zhang et al., 2010). Thus a semi-empirical correction
factor of $10^{-6}$ has been applied previously in PMCAMx-UF to scale the formation rates produced by the
Napari parameterization and better match the observations (Jung et al., 2010; Fountoukis et al., 2012;
Ahlm et al., 2013).

Encouraged by the good agreement between particle formation rates predicted by the ACDC model and
the state-of-the-art experimental data (Almeida et al., 2013), we have updated the particle formation
scheme within PMCAMx-UF with ACDC-based particle formation rates for the $NH_3$-$H_2SO_4$-$H_2O$ (see
Sect. 2.2 for details and the Results section for comparison to the scaled Napari parameterization). In
addition to applying the ternary $H_2SO_4$-$NH_3$-$H_2O$ NPF scheme, we also include a binary $H_2SO_4$-$H_2O$





NPF pathway. This pathway is operating simultaneously with the ternary pathway and is based on the
Vehkamäki et al. (2002) CNT-parameterization.

PMCAMx-UF was applied for the period of May 2008 for the European domain which consists of a
$5400 \times 5832$ km$^2$ region with a $36 \times 36$ km$^2$ grid resolution and 14 vertical layers reaching an altitude
of approximately 20 km. The PMCAMx-UF output data are hourly averaged. The meteorological inputs,
described in detail in Fountoukis et al. (2011; 2012), were created using the Weather Research and
Forecasting model version 2 (Skamarock et al., 2005) and include horizontal wind components, vertical
dispersion coefficients, temperature, pressure, water vapor mixing ratios, cloud optical depths and
rainfall rates. Hourly gridded emissions include anthropogenic emission rates of primary particulate
matter and gases. For the particle emissions the Pan-European anthropogenic Particle Number Emission
Inventory (Denier van der Gon et al., 2009; Kulmala et al., 2011) and the Pan-European Carbonaceous
Aerosol Inventory (Kulmala et al., 2011) were used. The anthropogenic gas emissions include both land
emissions from the GEMS data set (Visschedijk et al., 2007) and international shipping emissions. These
emission inputs are the same as have been used previously for the May 2008 period in PMCAMx-UF
(in Fountoukis et al., 2012; Ahlm et al., 2013), and thus in order to enable comparison to the previous
works these inputs are used in all of the base model runs of the present paper. To assess how much the
particle number concentrations are affected by emissions from natural sources we have performed
simulations with and without these emissions. The natural emissions include both particulate matter and
gases and combine three different datasets: emissions from ecosystems based on the MEGAN model
(Guenther et al., 2006), marine emissions based on the model of O'Dowd et al. (2008), and wildfire
emissions (Sofiev et al., 2008a, b).

**2.2 Improved treatment of the ternary NPF pathway**
The ternary H$_2$SO$_4$–NH$_3$–H$_2$O particle formation rate at approximately 1.2 nm in mass diameter was
calculated with the Atmospheric Cluster Dynamics Code (ACDC; Olenius et al., 2013; Almeida et al.,
2013; Henschel et al., 2015). ACDC simulates the dynamics of a population of molecular clusters by
numerically solving the cluster birth–death equations. Instead of considering only collisions and
evaporations of single vapor molecules, an often-used assumption applied in the CNT framework,
ACDC allows all possible collision and fragmentation processes within the cluster population. As input
the code needs the corresponding rate constants, of which the most challenging to assess are the cluster
evaporation rates, generally calculated from the free energies of formation of the clusters. The
evaporation rates play a significant role in determining the number concentration and consequently the
formation rate of small particles. The liquid drop model, commonly used in CNT to calculate the free
energies of cluster formation, is based on macroscopic thermodynamics and is thus not expected to give
reliable results for small clusters (Merikanto et al., 2007a). The most accurate theoretical method to



compute the free energies of clusters consisting of specific molecules is quantum chemistry. This
modeling approach is able to reproduce the general trends in cluster formation, and leads to, thus far,
the best quantitative agreement between observations and modeling with no fitting parameters (Almeida
et al., 2013).
In the ACDC simulations of this work, hard-sphere collision rates were used for the collision rate
coefficients, and the evaporation rate coefficients were calculated from the Gibbs free energies of
formation of the clusters computed with quantum chemical methods at the B3LYP/CBSB7//RICC2/aug-
cc-pV(T+d)Z level (Ortega et al., 2012; Henschel et al., 2014). This level of theory has been tested
against higher level methods and was shown to give reliable cluster formation free energies at an
affordable computational cost. The simulation included clusters containing up to three $H_2SO_4$ and three
$NH_3$ molecules, hydrated by up to four or five water molecules. Sulfuric acid and ammonia were
explicitly treated in the simulation, and water was implicitly included by assuming that the clusters are
in equilibrium with respect to water and by using hydrate averaged collision and evaporation rates. An
external sink term corresponding to scavenging by larger particles was used for all the clusters. The
steady-state particle formation rate was obtained as the flux of clusters growing out of the simulation
system considering boundary conditions based on cluster stability. Details of the simulated ternary
$H_2SO_4$–$NH_3$–$H_2O$ system can be found in Henschel et al. (2015).
The ACDC results were implemented  in the PMCAMx-UF framework as a look-up table consisting of
a comprehensive set of particle formation rates computed at different values of $H_2SO_4$ and $NH_3$
concentrations, temperature, RH, and coagulational loss rate due to scavenging by the population of
larger particles (described by the condensation sink, see e.g. Dal Maso et al., 2002). The formation rate
data produced by theoretical models have been traditionally fitted to a multivariable functional form
(Napari et al, 2002; Merikanto et al., 2007b), with the resulting parameterization then utilized by large
scale models. However, finding a suitable functional form to cover satisfactorily the whole parameter
space becomes increasingly difficult with increasing number of input parameters, with increasing
number of species participating in NPF, and with the tendency of formation rates to exhibit rapid, step
function–like changes with respect to one or more parameters. Thus interpolating from a look-up table
provides formation rates that are more closely in line with the original theoretical model, with a relatively
minor additional computational cost. The parameter space encompasses sulfuric acid concentration
between $1.00 \cdot 10^4$ and $3.16 \cdot 10^9$ molecules cm$^{-3}$, ammonia concentration between $10^6$ and $10^{11}$
molecules cm$^{-3}$, relative humidity between 0 and 100 %, temperature between 180 and 320 K and
condensation sink between $10^{-5}$ and $10^{-1}$ s$^{-1}$. These conditions bound the environmental and chemical
conditions predicted by typical PMCAMx-UF runs for Europe in May.  PMCAMx-UF uses multilinear
interpolation to extract formation rates from the look-up table. The newly-formed particles added to
PMCAMx-UF are assumed to have a diameter of 1.2 nm, corresponding to the size for which the ACDC





formation rates were calculated. This approach provides PMCAMx-UF with formation rates that are
based on the full kinetic treatment of the cluster population.

**2.3 Radiative transfer and photolysis rates**
Aerosols and clouds can enhance or reduce photolysis of relevant gas-phase chemical species in the
atmosphere by reflecting, scattering, or absorbing solar radiation. Modifications of photolysis rates via
this interaction lead to changes in the production rate of sulfuric acid, which lead directly to changes in
the new particle formation rates. Previous versions of PMCAMx-UF employed a parameterization
originally used by the Regional Acid Deposition Model (RADM; Chang et al., 1987) to treat the
modification of photolysis rates due to cloud presence. This approach required the cloud optical depth
from the meteorological input data and the solar zenith angle in order to calculate the time- and layer-
dependent adjustment factors for the photolysis rates. This method, however, did not use aerosol
concentrations predicted online by the transport model. Instead, a reference aerosol profile was used for
every time step and column of grid cells.

To more realistically treat the effects of clouds on the photolysis rates profile of the atmospheric column,
we updated the online approach in PMCAMx-UF to a streamlined form of the two-stream radiative
transfer module, TUV (Tropospheric Ultraviolet and Visible radiative transfer model; Madronich,
2002). The implementation of TUV was completed as documented by Emery et al. (2010). This
simplified module employs a reduced number of wavelength bands and plane-parallel two-stream
approximations. Inputs needed include the cloud optical depth, solar zenith angle, three-dimensional
aerosol concentration profile, and optical properties of the aerosol components provided by Takemura
et al., 2002.

The total cloud optical depth $\tau$ above a current grid cell up to the top of troposphere is here approximated
by

$$\tau = \frac{3L\Delta z_c}{2\rho_w r}, \qquad (1)$$

where $L$ is the mean cloud liquid water (g m$^{-3}$) , $\Delta z_c$ is the mean depth of cloudy layer (m) in the cell, $\rho$
is the density of water ($10^6$ g m$^{-3}$), and $r$ is the mean cloud drop radius ($10^{-5}$ m). The grid cells with cloud
optical depth less than 5 are considered as optically thin clouds (or cloud-free conditions), so that the
TUV module is not called for such grid cells. The module also takes as input the time- and space-
dependent vertical profile of dry and wet (with an RH-dependent lensing effect) aerosols predicted by
PMCAMx-UF.



The module outputs a modified actinic flux that can then be applied, using the clear-sky actinic flux for
reference, to adjust the clear-sky photolysis rates. Adjustments due to clouds and aerosols tend to reduce
photolysis below clouds but often enhance rates above clouds because of the reflection from the top of
the cloud. Emery et al. (2010) implemented the module in the Comprehensive Air Quality Model with
Extensions (CAMx) and evaluated it for ozone prediction in the Houston area. That study found
decreased ozone surface concentrations with maximum decreases of approximately 10 ppb. However,
they did not report the impacts that the radiation feedback would have on particulate mass or number.
We compare particle number and sulfuric acid vapor profiles with and without the radiation update in
place to better understand the importance of correctly representing this phenomenon.

**2.4 Model evaluation with particle number and size distribution data**
During the European Aerosol Cloud Climate and Air Quality Interactions (EUCAARI) project (Kulmala
et al., 2009; 2011) particle number size distributions within the atmospheric boundary layer were
measured at various European Supersites for Atmospheric Aerosol Research (EUSAAR). May 2008
was one of the intensive observation periods of the project. In this study the predicted ground-level
hourly-averaged particle number concentrations are evaluated against the data from Aspvreten
(Sweden), Cabauw (Netherlands), Hyytiälä (Finland), Ispra (Italy), Mace Head (Ireland), Melpitz
(Germany) and Vavihill (Sweden) similarly to Fountoukis et al. (2012). These locations represent seven
different types of European environments (Ahlm et al., 2013). More information about the
characteristics and topography of these sites is available elsewhere (Asmi et al., 2011 and Fountoukis et
al., 2012). The particle size distribution measurements were carried out using either a Differential
Mobility Particle Sizer (DMPS) or Scanning Mobility Particle Sizer (SMPS) systems in the mobility
diameter size range above 10 nm.
To evaluate the vertical profile of the particle size distribution, we used the observational data measured
by the German DLR Falcon 20 and the British FAAM BAe-146 research aircrafts, operating between 6
and 24 May 2008. The aircraft data was collected during the LONGREX campaign (Hamburger et al.,
2011), which was also a part of the EUCAARI project. The FAAM BAe-146 flights mainly flew in the
boundary layer and lower free troposphere while the DLR Falcon 20 aircraft mostly probed the free
troposphere up to the tropopause level (Hamburger et al., 2011). The Condensation Particle Size
Analyser (CPSA) (Fiebig et al., 2005; Feldpausch et al., 2006), installed aboard the DLR Falcon 20, and
the Passive Cavity Aerosol Spectrometer Probe (PCASP-100X) (Liu et al., 1992), operated aboard both
aircraft, measured the particle number concentrations. Consistent with Reddington et al. (2011), we used
the measurements from two channels of the CPSA onboard the DLR Falcon 20 with lower cut-off
diameters of 4 and 10 nm, yielding the number concentrations of particles above these sizes, denoted as
$N_4$ and $N_{10}$. The temporal resolution of the CPSA dataset is 1 s. The nominal size range of PCASP-100X
is 0.12-3.5 μm with 15 channels. The PCASP-100X raw data was sampled with 1 Hz frequency, but the



data used here is based on averaging over a constant interval of 5 s. We used the measured particle
number concentrations obtained from channels 3 to 10 of the PCASP-100X covering the diameter range
of 160-1040 nm, representative of accumulation mode, also to facilitate comparisons with the results
reported by Reddington et al. (2011). We also used measurements by a TSI 3786 Condensational Particle
Counter (CPC) aboard the FAAM BAe-146 aircraft measuring the number concentrations of particles
larger than 4 nm.
A map of flight tracks by the Falcon 20 and Bae-146 and more details about EUCAARI-LONGREX
dataset is available elsewhere (Reddington et al., 2011; Hamburger et al., 2012). Measurements from
the LONGREX campaign span altitudes corresponding to 13 of the 14 vertical layers of PMCAMx-UF
(Fig. S1 in the supplement). The model data were paired with the aircraft data by converting the time-
dependent latitude, longitude, and altitude of the plane to a model grid-cell index.

## 299    3 Results

### 300    3.1 Surface-level particle number concentrations

In this study we explore the sensitivity of PMCAMx-UF to (1) updated NPF scheme with ACDC-based
formation rates, (2) updated cloud adjustment scheme with TUV implementation, and (3) including
natural particle number emissions. The baseline simulation (hereafter ACDC-TUV-DE; see Table 1)
represents a prediction of the particle number concentrations with implementation of ACDC-based NPF
scheme and TUV cloud adjustment scheme while using the default (i.e. only anthropogenic) particle
emissions similarly to Fountoukis et al. (2012). Table 1 summarizes the simulations reported in this
study. Figure 1 shows the arithmetic mean number concentration over May 2008 at ground-level for
each PMCAMx-UF grid cell for particles larger than 10 ($N_{10}$), 50 ($N_{50}$) and 100 ($N_{100}$) nm and all particles
($N_{tot}$) as predicted using the baseline simulation ACDC-TUV-DE. The first two days of the simulation
were excluded from the analysis to minimize the impact of the initial conditions on the results. The
domain mean during May 2008 for $N_{tot}$ is 59200 cm$^{-3}$, for $N_{10}$ the corresponding number is 7100 cm$^{-3}$,
for $N_{50}$ 1300 cm$^{-3}$, and for $N_{100}$ 360 cm$^{-3}$. The spatial pattern of the predicted number concentrations is
similar to the results reported by Fountoukis et al. (2012), which were obtained using the simulation
Napari-RADM-DE. The highest number concentrations are predicted over Eastern Europe during this
photochemically active period while the lowest particle number concentrations are predicted over
Nordic countries. The simulation Napari-TUV-DE predicts the domain mean of $N_{tot}$, $N_{10}$, $N_{50}$ and $N_{100}$
of 8100, 4000, 1500 and 410, respectively. Although updating the NPF scheme of PMCAMx-UF with
ACDC-based formation rates significantly affects the number of small particles with diameter below 10
nm, the spatial concentration remains unchanged. Updating the model cloudiness scheme by



implementing the TUV radiative transfer module did not greatly affect the spatial distribution of number
concentrations but caused a minor change in the number concentration values. This is confirmed by the
arithmetic domain mean values during May 2008 of $N_{tot}$, $N_{10}$, $N_{50}$ and $N_{100}$ predicted by the ACDC-
RADM-DE simulation, which are 62000, 6800, 1200 and 340 $cm^{-3}$, respectively, and thus very similar
to the baseline simulation. Including the natural particle emissions (in simulation ACDC-TUV-NE)
resulted in 48300, 6200, 1300 and 380 $cm^{-3}$ for $N_{tot}$, $N_{10}$, $N_{50}$ and $N_{100}$, respectively, therefore predicting
lower number concentrations of small particles (i.e. diameter < 10 nm) compared to that predicted by
the baseline simulation. This is probably due to the higher sink of newly formed particles caused by the
added natural particle emissions.

Figure 2 shows scatter plots of the predicted (ACDC-RADM-DE) versus measured hourly-averaged
$N_{10}$, $N_{50}$ and $N_{100}$ at the seven EUSAAR measurement sites during May 2008. More than 70 % of the
data points for $N_{50}$ and $N_{100}$ predictions fall generally within a factor of two of the measurements, with
slight underpredictions for $N_{100}$ (38% below the 2:1 line) at some sites, similar to the results predicted
by Napari-RADM-DE simulation reported by Fountoukis et al. (2012). However, the model using the
ACDC-based formation rates is overpredicting $N_{10}$, in particular over clean locations including
Aspvreten, Hyytiälä, Vavihill and Mace Head. The reason for this overprediction is most likely linked
to the missing condensable vapors and particle growth mechanisms in the simulations reported here (see
Fountoukis et al., 2012; Ahlm et al., 2013; Patoulias et al., 2015). When implemented, the additional
growth would likely increase the condensation sink by shifting the size distribution towards larger sizes.
However, given the fact that no empirical fitting parameters have been applied to the ACDC-based NPF
description, we deem the agreement encouraging. The biases presented here and in the following figures
can thus be considered conservative estimates. Furthermore, in this study we have only considered the
ternary sulfuric acid - water - ammonia particle formation scheme. There may be other significant
mechanisms present, e.g. sulfuric acid - amine particle formation (Bergman et al., 2015), with a
geographical pattern resembling that of our results. Both mechanisms depend on sulfuric acid
concentration, the model prediction of which can naturally be inaccurate as well. We compared the
modeled and measured acid concentrations at one of the measurement sites (Melpitz), and found that
the modeled concentrations were slightly overpredicted (Fig. S2 in the supplement). This may also
contribute to the overprediction of the small particle sizes.

**3.2 Vertical profiles of particle number concentrations**
In this section we investigate the vertical distribution of the means of $N_{tot}$, $N_{10}$, $N_{50}$ and $N_{100}$ along with
parameters relevant for predicting NPF for the base case simulations (Fig. 3). These parameters include
gas-phase concentrations of $H_2SO_4$ and $NH_3$, RH and $T$. In the results shown in Fig 3 the TUV radiation
scheme has been used, thus representing the baseline simulation ACDC-TUV-DE. As can be seen from





Fig. 3, particles smaller than 10 nm contribute significantly to the total number concentration throughout
the tropospheric column, $N_{tot}$ is about one order of magnitude greater than $N_{10}$ and two and three orders
of magnitudes greater than $N_{50}$ and $N_{100}$, respectively. Values of $N_{10}$, $N_{50}$ and $N_{100}$ decrease monotonically
with altitude, dropping significantly at approximately 1 km (layers 6-8 of the model). The vertical
distribution of $N_{tot}$ shows a different trend at higher altitudes where a bump in $N_{tot}$ occurs at around 6-
11 km, although no significant increase in the gas phase concentration of $H_2SO_4$ and $NH_3$ are predicted
at these altitudes (Fig. 3). The increase in $N_{tot}$ is mostly due to significantly decreased coagulation sink
for the newly-formed particles, as the number of larger particles dramatically decreases with altitude,
and partly due to the rapidly decreasing temperature. PMCAMx-UF predicts the particle formation rates
to decrease rapidly from around 2 km upward. The temperature, RH and sulfuric acid profile have
similar relative trends as the $N_{10}$, $N_{50}$ and $N_{100}$ profiles. There is a plateau in temperature and RH (at
temperature range 285-288 K and RH range 80-83 %) profile up to altitude 1.2 km. Above this altitude,
however, the RH and temperature values decrease rapidly. The sharp decreases in the relative humidity,
temperature and particle number concentrations are consistent with the location of the boundary layer
height. This is in agreement with Ferrero et al., (2010) who showed that mixing height estimations (over
the city of Milan) derived from particle number concentration, temperature and relative humidity are
correlated with one another.
Figure 4 shows the comparison of the two simulations ACDC-TUV-DE and Napari-TUV-DE (see Table
1) with the observational data collected during the EUCAARI-LONGREX campaign measured by
German DLR Falcon 20 and the British FAAM BAe-146 aircraft. The model using the ACDC-based
formation rates predicts the number concentration profile of particles larger than 4 nm ($N_4$) within about
one order of magnitude of the observed $N_4$ profile throughout the atmospheric column. The scaled
Napari NPF scheme leads to $N_4$ concentrations somewhat closer to the observations than those using the
ACDC scheme. As mentioned above, the vertical profiles presented in Figure 4 are produced by the
model using the TUV radiation scheme. A similar analysis of the vertical profiles using the RADM
radiation scheme (simulation ACDC-RADM-DE), which is not shown here, results in exactly the same
shape of the number concentration profiles. The vertical profiles using the RADM radiation scheme
show negligible difference in the absolute number concentrations with slightly worse agreement with
the observations compared to the TUV radiation scheme. The number concentrations of particles larger
than 10 nm ($N_{10}$) predicted by the model using the scaled Napari NPF scheme agrees reasonably well
with the observations throughout the atmospheric column. The model using the ACDC formation rates
tends to slightly overpredict the $N_{10}$ profile. The shape of the observed $N_{10}$ vertical profile is captured
reasonably well throughout the atmospheric column regardless of the NPF scheme used. Both model
versions have almost the same performance for the $N_{160-1040}$ profile within the boundary layer; both
simulations (i.e. ACDC-TUV-DE and Napari-TUV-DE) underpredicting the $N_{160-1040}$ profile by about
half an order of magnitude. This behavior is seen in the $N_{160-1040}$ profile corresponding to both



observational data sets (i.e. Falcon 20 (Fig. 4-d) and BAe 146 (Fig 4-e) aircraft data). This is at least
partly due to the lack of sources of organic condensable vapors to grow the particles to larger sizes in
the model (Patoulias et al., 2015), which will be investigated in a future study. The underprediction
decreases for all model versions at altitudes above the boundary layer improving the agreement with
observational data.
The results for the model using the ACDC-based formation rates are comparable to previous studies.
For example, Reddington et al. (2011) tested different NPF parameterizations in BL including activation,
kinetic and combined organic-$H_2SO_4$ parameterizations, which are implemented in the Global Model of
Aerosol Processes (GLOMAP). The evaluation of the modeled vertical profiles of particle number
concentrations against the aircraft measurements similar to this study showed that all of the mentioned
NPF schemes dramatically under-predicted particles in nucleation (normalized mean bias (NMB) varies
from -33 to -96) and Aitken mode sizes (-44 < NMB < -59). The larger particles ($N_{100}$) however were
generally well-captured by the model. Furthermore, Lupascu et al. (2015) compared simulated number
concentrations with aircraft measurements collected during the Carbonaceous Aerosol and Radiative
Effects Study (CARES) campaign. They also tested different NPF parameterizations including
activation, kinetic and combined organic-$H_2SO_4$ parameterizations, which are implemented in the
regional scale model WRF-Chem one-at-a-time using a sectional framework to simulate the NPF. They
found that their simulations overpredicted the particle number concentrations, especially in the smallest
sizes (normalized mean bias of 126-608 % for $N_3$ and $N_{10}$). The nucleation scheme had very little impact
on the magnitude of the CCN-sized particle number concentrations.

### 3.3 Effect of the radiative transfer scheme on predictions of particle number concentrations

Updating the radiative transfer scheme to the TUV scheme has a small effect on the predicted number
concentrations; the vertical profile of the relative difference ($N_{TUV}$ - $N_{RADM}$) × 100 / $N_{RADM}$ in the May-
2008 domain mean particle number concentrations shows a maximum reduction of about -5.5 % in $N_{tot}$
(at altitude 2.2 km) and a maximum increase of about 9 % in $N_{100}$ (at altitude range 0.7-2.2 km). Figure
5 shows the spatial distribution of the absolute difference of the $H_2SO_4$ gas phase concentration and total
particle number concentrations between the simulations ACDC-TUV-DE and ACDC-RADM-DE  (see
Table 1) at 12:00 UTC on May 5, 2008. Figure 5 also presents the cloud optical depth fields to illustrate
the link between the cloud fields and changes in the particle number concentrations due to the new cloud
adjustment scheme. The TUV scheme results in higher particle formation rates above and in the vicinity
of the cloudy regions due to enhanced radiation and sulfuric acid production. This is in agreement with
observations reported by Wehner et al. (2015). They concluded that the cloudy regions provide a
favorable environment for NPF above and at the edges of clouds due to enhanced upward spectral
irradiance and cloud-reflected spectral radiance around them. Sulfuric acid concentration is reduced




below cloud in the TUV scheme, due to the enhanced UV attenuation scaling down the photolysis rates.
However, as pointed out above, the effect on the total particle number concentrations is generally small.

## 4 Conclusions


We have updated the new particle formation (NPF) scheme within PMCAMx-UF with particle
formation rates for the ternary $H_2SO_4$-$NH_3$-$H_2O$ pathway simulated by the Atmospheric Cluster
Dynamics Code using quantum chemical input data. The ACDC results were implemented in
PMCAMx-UF as a lookup table from which the formation rates were interpolated. We believe this is
the first time that reasonable particle concentrations have been produced in a large-scale atmospheric
model with a NPF scheme without any scaling factors or location/condition dependent semi-empiricism.
In addition to the updated NPF description, we have also updated PMCAMx-UF treatment of the
cloudiness effect on the photolysis rates  (i.e. cloud adjustment scheme) profile by implementing a
streamlined version of the Tropospheric Ultraviolet and Visible radiative transfer model (Madronich,

439    2002).

We used the updated PMCAMx-UF to simulate particle number concentration during May 2008 over
Europe. During this period, the EUCAARI campaign was performed to measure the particle number
size distributions within the atmospheric boundary layer at various European Supersites for Atmospheric
Aerosol Research (EUSAAR) in addition to higher altitude data collected by two research aircraft during
the LONGREX campaign. Comparing the measured particle number concentrations at the EUSAAR
sites to the predictions of the updated PMCAMx-UF shows that the model slightly overpredicts
concentrations for particles with diameters between 10-100 nm. Particles larger than 100 nm are slightly
underpredicted. In general, the model predictions of number concentrations of aerosols, in particular
particles within Aitken and accumulation mode sizes agree reasonably well with the measurements.
Vertical profiles of particle number concentrations show that predicted concentrations of small particles
are within one order of magnitude of the aircraft measurements. The predicted Aitken- and accumulation
mode number concentrations are in quite good agreement with the observational data throughout the
atmospheric column, while the concentrations of smaller particles are somewhat overpredicted by the
ACDC-based NPF scheme. Including organic condensation onto the ultrafine particles could improve
these predictions.
Overall, we consider our results very promising: a NPF scheme based on first-principles theory and no
artificial scaling is shown to be a promising alternative to semi-empirical approaches in the description
of particle formation in large scale atmospheric models.



## Code availability


The look-up table used for the representation of particle formation rates is openly available for download
at http://www.aces.su.se/research/research-facilities/models. The updated NPF and TUV modules are
available from I. Riipinen (ilona.riipinen@aces.su.se).

## Acknowledgements


We gratefully acknowledge Oona Kupiainen-Määttä for providing the ACDC-simulation data and
generating the look-up table. Tinja Olenius is acknowledged for discussions and technical support
related to ACDC, and Radovan Krejci for providing the EUCAARI-LONGREX data. Falcon
measurements and data analysis were funded by EUCAARI (European Integrated project on Aerosol
Cloud Climate and Air Quality interactions) project No. 036833-2 and by DLR. The UK aircraft
experiment was supported through EUCAARI and the UK Natural Environment Research Council
through the APPRAISE programme, grant NE/E01108X/1. The authors also thank the Academy of
Finland Center of Excellence program (project number 272041), the Nordic Centre of Excellence
CRAICC , Academy of Finland, ERC-StG-ATMOGAIN (278277) and ERC-StG_MOCAPAF

472 (257360).

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





## Figures and figure captions

Table 1. Summary of PMCAMx-UF model simulations reported in this study. The arithmetic mean of
ground-level number concentration during May 2008 for particles larger than 0.8 nm ($N_{tot}$), 50 nm ($N_{50}$)
and 100 nm ($N_{100}$) is given for each simulation.  DE = default emissions, NE = new emissions. The
"default emissions" refer to the emissions used in Fountoukis et al., 2012 (simulation Napari-RADM-
DE).

| Simulation name | NPF scheme | Cloud adjustment scheme | Emissions | Domain mean number concentration (cm$^{-3}$) | | |
|---|---|---|---|---|---|---|
| | | | | $N_{tot}$ | $N_{50}$ | $N_{100}$ |
| ACDC-TUV-DE | ACDC-based | TUV | Default | 59200 | 1300 | 360 |
| ACDC-RADM-DE | ACDC-based | RADM | Default | 62000 | 1200 | 340 |
| ACDC-TUV-NE | ACDC-based | TUV | Updated | 48300 | 1300 | 380 |
| Napari-TUV-DE | Scaled Napari et al., 2002 | TUV | Default | 8100 | 1500 | 410 |
| Napari-RADM-DE | Scaled Napari et al., 2002 | RADM | Default | 9000 | 1500 | 400 |







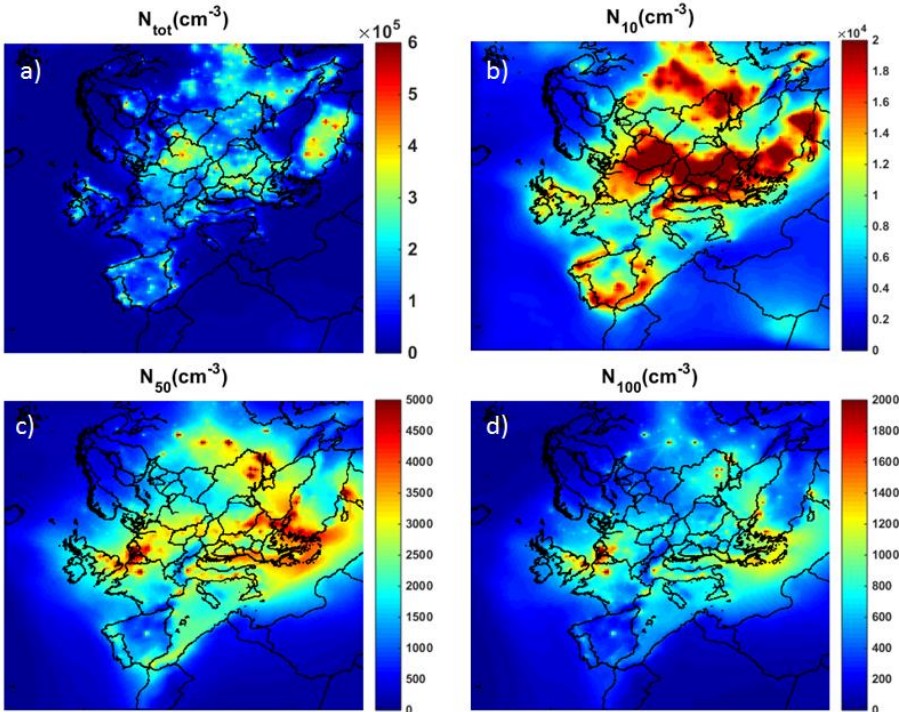


Figure 1. The simulated spatial distribution of the arithmetic mean of ground-level number concentration during May 2008 for particles larger than (a) 0.8 nm ($N_{tot}$), (b) 10 nm ($N_{10}$), (c) 50 nm ($N_{50}$), and (d) 100 nm ($N_{100}$). The PMCAMx-UF baseline simulation ACDC-TUV-DE is used (see Table 1). Note that different color bar scales are used for the different size ranges for readability.




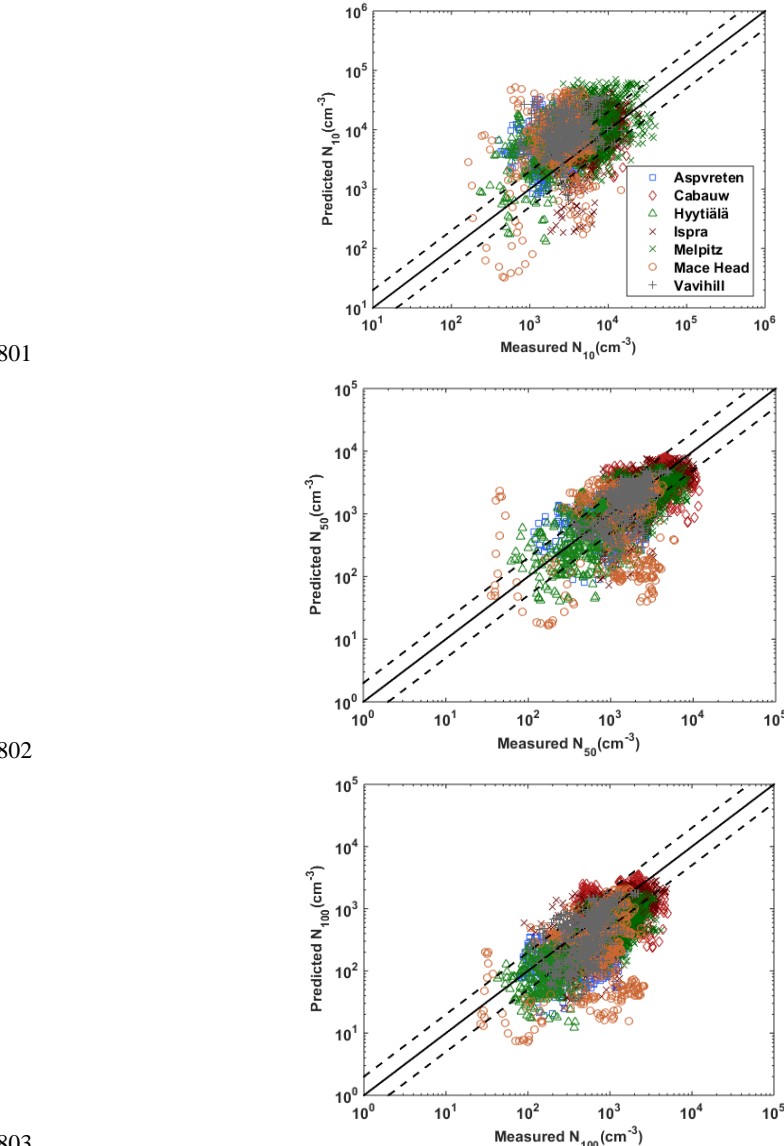




Figure 2. Comparison of predicted vs. measured hourly-averaged number concentration of particles larger than 10 nm ($N_{10}$), 50 nm ($N_{50}$) and 100 nm ($N_{100}$) during May 2008 from the 7 EUSAAR measurement stations during the EUCAARI project. Lines corresponding to 1:1 (solid line), and 1:2 and 2:1 (dashed lines) are shown. The PMCAMx-UF model simulation ACDC-RADM-DE is used (see Table 1).






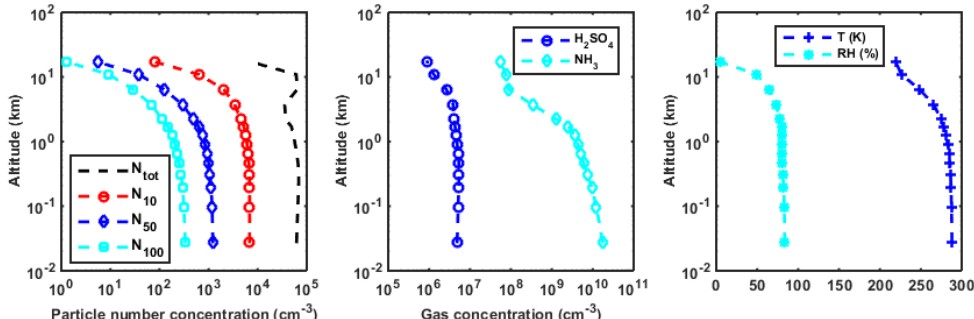


Figure 3. Vertical profiles of simulated variables averaged (arithmetic mean) over May 2008 and the whole simulation domain. Left panel: number concentration (cm$^{-3}$) of particles larger than 0.8 nm ($N_{tot}$), 10 nm ($N_{10}$), 50 nm ($N_{50}$) and 100 nm ($N_{100}$). Middle panel: gas phase concentration (cm$^{-3}$) of sulfuric acid (H$_2$SO$_4$) and ammonia (NH$_3$). Right panel: temperature (K) and relative humidity (%). The PMCAMx-UF baseline simulation ACDC-TUV-DE is used (see Table 1).







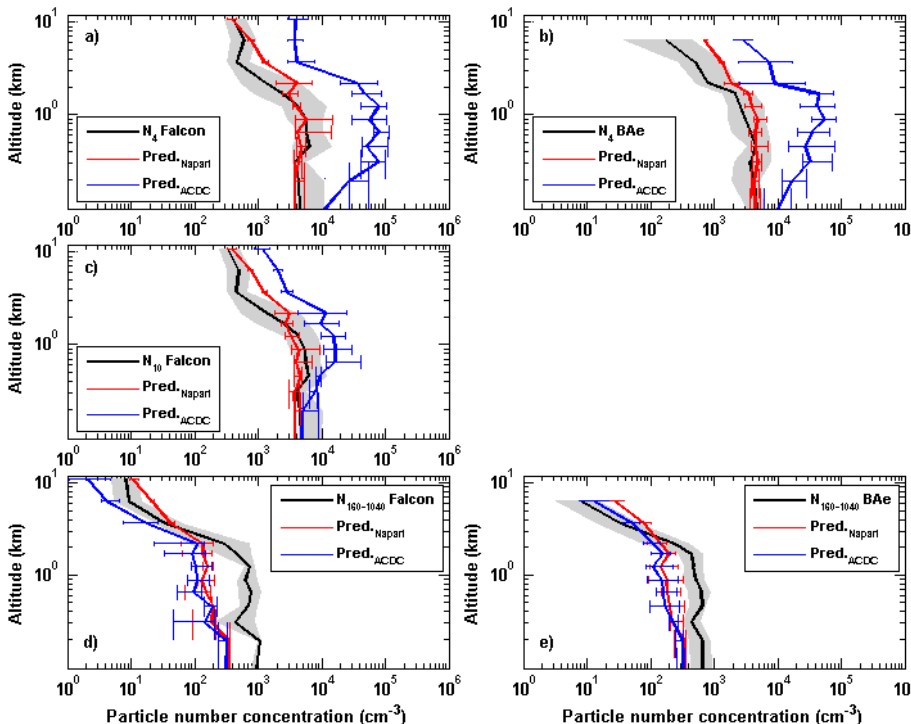


Figure 4. Vertical profiles of measured (black) and predicted (red and blue) particle number
concentrations for the size ranges: (a) and (b) Larger than 4 nm ($N_4$) measurements collected by Falcon
and BAe 146, respectively, (c) larger than 10 nm ($N_{10}$) measurements collected by Falcon 20, (d) and
(e) 160-1040 nm ($N_{160-1040}$) measurements collected by Falcon and BAe 146, respectively, during May
2008. Red and blue lines show the predicted particle number concentrations by the PMCAMx-UF model
using ACDC-based formation rates (ACDC-TUV-DE) and scaled Napari new particle formation scheme
(Napari-TUV-DE), respectively. The lines show the median values of data points within each model
layer, and the error bars and grey shading indicate the values between 25-th and 75-th percentiles of the
model results and observations, respectively. Concentrations are given at ambient temperature and
pressure.


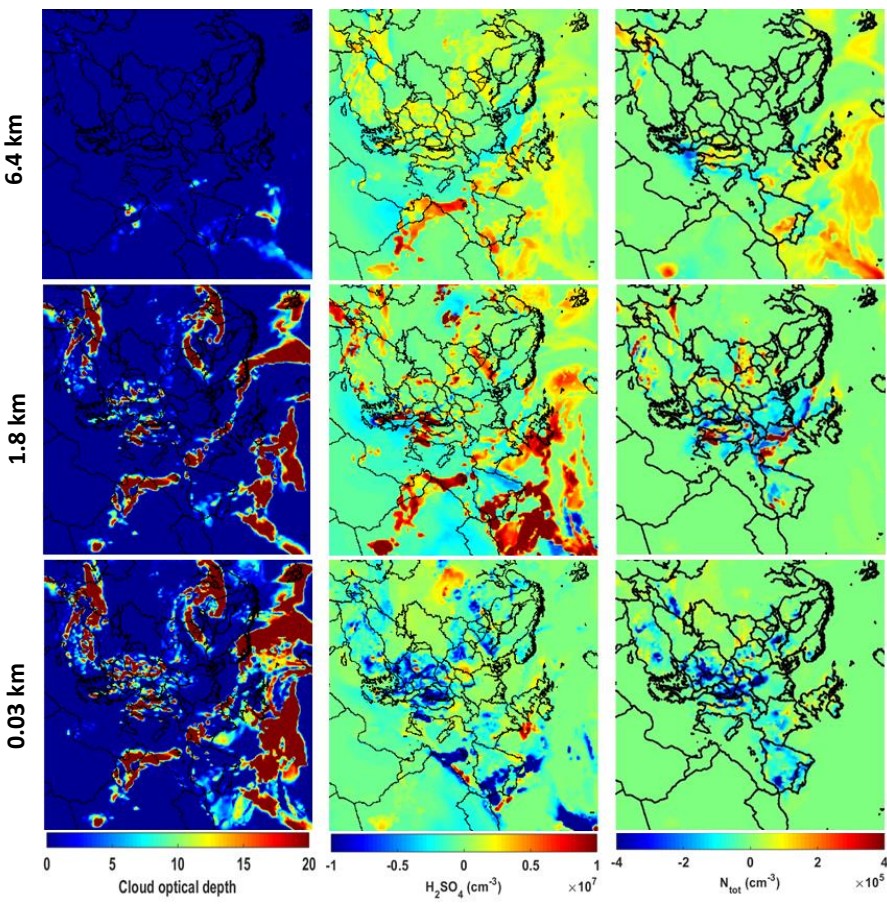


Figure 5. Left column: the total cloud optical depth supplied by WRF meteorology model. Middle
column: the absolute difference between the predictions using the TUV (the simulation ACDC-TUV-
DE; see table 1) and RADM (the simulation ACDC-RADM-DE) radiative transfer schemes within
PMCAMx-UF for $H_2SO_4$ concentration. Right column: absolute difference between prediction using
TUV and RADM schemes for total particle number concentrations $N_{tot}$. The parameters shown in the
figure are snapshots on May 5, 2008 12:00 UTC at model layers 1 (mid-point altitude 0.03 km), 9 (mid-
point altitude 1.7 km) and 12 (mid-point altitude 6.4 km).