# Peer review of "Implementation of state-of-the-art ternary new particle formation scheme"

_Geoscientific Model Development, 2016_

## Referee Comment (RC1) · J.R. Pierce (Referee) · 13 Feb 2016

This paper documents the implementation of a new nucleation scheme, based on the ACDC model, into a chemical transport model, PMCAMx-UF over Europe. Also, sensitivities to the radiation scheme and natural emissions were performed. The paper is in review for GMD and has a focus on model development rather than science findings. I feel like the scope of the paper is appropriate for the journal, and the overall research approach is good. However, the paper makes claims that I feel are unsubstantiated, lacks quantitative evaluation in many places, and is unclear in some places. Thus, I feel that the paper needs some substantial revisions before publication.

Broad comments:
[Figure]

1) Lack of quantification against measurements: Figure 2 shows comparison of a single simulation (ACDC-RADM-DE... not ACDC-TUV-DE, which seemed to be the default for the other figures). While a few numbers are given for the percentage of points that fall within a factor of 2 of measurements, there aren't other statistics for slope and bias (N10 seems to be biased 5-10x too high on average). Further, why aren't statistics given for the other simulations? It would be straightforward to quantitatively compare the simulations against the measurements in this way, and this would show us which assumptions moved the model closer to measurements.

Similarly, this can be done for the vertical profiles. N4 are more than 10x too high with the ACDC scheme throughout most (∼70%) of the boundary layer, and N10 is more than a factor of 2 too high for half of the the boundary layer. Please give some summary statistics.

2) Unsubstantiated claims: The paper concludes with, "Overall, we consider our results very promising: a NPF scheme based on first-principles theory and no artificial scaling is shown to be a promising alternative to semi-empirical approaches in the description of particle formation in large scale atmospheric models." There is similar discussion earlier in the conclusions, "reasonable particle concentrations". Obviously, "very promising" and "reasonable" are subjective statements, and different people would view the 10x boundary-layer overprediction for N4 as not "very promising" or "reasonable". We can use Westervelt et al. (2014) for comparison, where they compare simulations using scaled Napari (1E-5 in this case) versus unscaled Napari (also "first principles", and in the unscaled case, without a tuning factor). Table 3 and Figure 2 in Westervelt shows that N10 is only 2x higher in the boundary layer for the unscaled Napari simulation relative to the scaled Napari simulation... similar to what was found in the current manuscript in review between ACDC and scaled Napari (1E-6 in the current manuscript). Thus, the overpredictions by the ACDC scheme relative to the scaled Napari in this current manuscript is not unlike comparing scaled and unscaled Napari. I would guess that the nucleation rates predicted by ACDC must be many orders of

magnitude faster than scaled Napari in order to get N4 to be 10x too high given all of the microphysical dampening between nucleation rates and N4.

While I know that Napari cannot be right for the right reasons, it is very surprising that ACDC does so poorly here considering that it does ok relative to CLOUD measurements (is there a mistake in the lookup tables? are these formation rates per mˆ3 rather than per cmˆ3?). What are the mean nucleation rates predicted at some sites where nucleation rates have been well-characterized (e.g. Hyytiala)?

I agree that missing condensible material may be part of the issue here (adding more would likely lower N4 and increase N100). D'Andrea et al. (2013) lowered N10 by about 20-50% over Europe by adding 100 Tg yr-1 globally of non-volatile SOA correlated with anthropogenic CO (see Figure 3). Thus, the N10 bias in the current manuscript may be somewhat fixed by missing condensible material. But regardless, please don't make unsubstantiated claims about the current setup since the ACDC rates seem to be similar to the unscaled Napari scheme that we love to hate.

Specific comments:

L62: Adams and Seinfeld (2002) did not use the Napari nucleation scheme.

L145: Why not use ACDC with no ammonia for binary nucleation?

L160-165 (as well as throughout): These lines come out of nowhere but seem important. Were natural sources not included in PMCAMx-UF before? I didn't figure this out until later in the paper. While there is a test simulation turning natural emissions on, there is never much discussion about this in the paper. Why isn't having natural emissions on the default setting in this paper (with a sensitivity simulation with them off). I realize you chose this because you didn't have natural emissions before, but shouldn't the simulations with natural emissions be expected to better simulate the size distribution? Please give attention to the text and discussion of natural emissions throughout to ensure that it is clear.

L246-247 (and this entire paragraph): Why is TUV *not* called for cells with clouds thinner than optical depth 5? Don't you want to consider radiative transfer here? Earlier you discuss wanting to have the effects of simulated aerosols on photolysis rates... so wouldn't you want this for clear-sky or thin-cloud conditions? Confusing.

L319: "spatial concentration". Do you mean the relative spatial distribution of concentrations? Since you don't show figures for these other simulations, would it make sense to quantify this in some way, e.g. a correlation coefficient between simulations?

Figures: Why is the default simulation shown in figures sometimes ACDC-TUV-DE (most figures) and sometimes ACDC-RADM-DE (Figure 2)?

Figures 3 and 4: Why is the height axis log scale? Do we want to focus mostly on the boundary layer?

L377-379: "somewhat closer"? Within 1.5x for scaled Napari vs. outside of 10x for ACDC for most of the boundary layer?! Please be quantitative and avoid subjective judgement.

General grammar comment: Adverbs modifying an adjective do not need a hyphen (and should not have a hyphen), e.g. "vertically resolved", "chemically resolved", "newly formed", "hourly averaged". There is no ambiguity in the meaning with or without a hyphen. These should be removed.

On the other hand, it is extremely useful to hyphenate joint adjectives. For example, "Aitken-mode particles" should have a hyphen as they are particles in the "Aitken mode". There are not "mode particles" that are "Aitken". Same with "chemical-transport models". They are not "transport models" that are "chemical". How about "large scale atmospheric models"? What is a "scale atmospheric model", and what is so large about it? It is commonplace, unfortunately in my opinion, to omit hyphens when the writer thinks the meaning is unambiguous. However, if a writer never hyphenates joint adjectives, we get into trouble when the writer writes, "slow moving van". Is it a "moving

van" that is going slow? Or is it a regular van that is "slow moving"? If a writer establishes that they will hyphenate whenever it is appropriate, we would know that "slow moving van" means a "moving van" that is going slow (else they would have written "slow-moving van").

---

## Referee Comment (RC2) · Anonymous Referee #2 · 2 Apr 2016

This manuscript presents the implementation of a new ternary $H_2SO_4$-$H_2O$-$NH_3$ parameterization, into the PMCAMx-UF model. The authors explore the ability of the model to reproduce observed number concentration during May 2008, when the intensive observation period of EUCAARI project took place. Apart from the testing of the new parameterization, sensitivity tests using the scaled Napari parameterization and sensitivity to the radiation scheme and natural emissions were performed. The topic and overall approach fits with GMD; therefore, I am in favor of accepting this work for publication in GMD after the authors have addressed the issues summarized below.

Major issue:

While several sensitivity tests are done, the paper lacks a proper statistics for each test. Figure 2 shows the results coming from ACDC-RADM-DE sensitivity study, however no information is given for the other studies presented in Table 1. It would be nice to see some numbers (r, over/under estimation factor, bias), to endorse the statement "Overall, we consider our results very promising: a NPF scheme based on first-principles theory and no artificial scaling is shown to be a promising alternative to semi-empirical approaches in the description of particle formation in large scale atmospheric models."

Specific comments:

L82: Should be "Matsui et al., 2011, 2013", not "Matsui et al., 2011a, 2013c".

L88-90: Matsui et al., 2013 study, already mentioned by the authors, have also assessed the ability of WRF-Chem to reproduce the vertical profile of observed Aitken particles for South Asia.

L133-134: Should be "Yu et al., 2006", not "Yu et al., 2006a".

L210-213: I assume that if the $H_2SO_4$, $NH_3$, RH, temperature and condensation sink are not falling into the mentioned range, the Vehkamaki et al., 2002, parameterization is applied. Is that right?

L330-336: The authors show the scatter plots of predicted PNC using ACDC-RADM-DE simulation vs observed PNC in several size ranges. Yet, at line 303 they state that ACDC-TUV-DE is the baseline simulation. Do they have any particular reason not to present the results coming from the default simulation? As can be seen in Table 1 the differences between the ACDC-RADM-DE and ACDC-TUV-DE simulations are minors. Furthermore, they use the ACDC-TUV-DE simulation results for the following plots. A little bit confusing.

L373-379: Could you give an explanation why N4 concentration increases in the upper boundary layer for ACDC-TUV-DE simulation? May you could present the particle formation rates for the ACDC-TUV-DE and Napari-TUV-DE simulation. Also, could you give an overestimation factor?

L384-389: An index of agreement will sustain "the scaled Napari NPF scheme agrees reasonably well with the observations throughout the atmospheric column" and "reasonably well" statements.

L433_435: The following sentence for a more scientifically sound expression  should be rephrased: "We believe this is the first time that reasonable particle concentrations have been produced in a large-scale atmospheric model with a NPF scheme without any scaling factors or location/condition dependent semi-empiricism".

Conclusion section: The authors should be more restrictive in using "reasonably well", "are somewhat overpredicted by the ACDC-based NPF scheme", "very promising" statements due to the fact that the lack of statistics throughout the paper does not sustain their claims.

---

## Author Response (AR1)

**Dear editor,**

We thank both reviewers for their insightful comments on our work. The point-by-point responses to all issues raised can be found below. We have also revised the manuscript accordingly. The manuscript revised according to the reviewer suggestions can be found at the end of this letter. In the revision process, we have also added one more co-author, Dr. Lars Ahlm, to the revised manuscript. Furthermore, we have corrected a few typos and the map projections in Fig. 5.

On behalf of the co-authors,

Ilona Riipinen

**Prof. Jeffrey Pierce (Reviewer #1)**

We thank Prof. Pierce for his time as well as the comments and suggestions for our manuscript. Our point-bypoint responses for the issues raised can be found below. The direct quotations from the comments are shown in italics, and our responses with normal font type.

This paper documents the implementation of a new nucleation scheme, based on the ACDC model, into a chemical transport model, PMCAMx-UF over Europe. Also, sensitivities to the radiation scheme and natural emissions were performed. The paper is in review for GMD and has a focus on model development rather than science findings. I feel like the scope of the paper is appropriate for the journal, and the overall research approach is good. However, the paper makes claims that I feel are unsubstantiated, lacks quantitative evaluation in many places, and is unclear in some places. Thus, I feel that the paper needs some substantial revisions before publication.

Thank you for these constructive comments. We hope that you find that our point-by-point responses below address the concerns raised. We will naturally revise the manuscript accordingly.

1) Lack of quantification against measurements: Figure 2 shows comparison of a single simulation (ACDC-RADM-DE... not ACDC-TUV-DE, which seemed to be the default for the other figures). While a few numbers are given for the percentage of points that fall within a factor of 2 of measurements, there aren't other statistics for slope and bias (N10 seems to be biased 5-10x too high on average). Further, why aren't statistics given for the other simulations? It would be straightforward to quantitatively compare the simulations against the measurements in this way, and this would show us which assumptions moved the model closer to measurements. Similarly, this can be done for the vertical profiles. N4 are more than 10x too high with the ACDC scheme throughout most (~70%) of the boundary layer, and N10 is more than a factor of 2 too high for half of the the boundary layer. Please give some summary statistics.

We originally decided to leave these statistics out since the changes in the model outputs were (as expected) small as compared with previous versions with less sophisticated NPF descriptions. To illustrate this point, the prediction skill metrics of the simulation ACDC-TUV-DE in the original manuscript are now summarized in Table R1 below, which can be compared to the similar results shown in Table 2 of Fountoukis et al., 2012. Table R2 shows the same metrics for the vertical profiles. It is clear that while the model skill in predicting the number concentrations of particles above 50 and 100 nm is similar as in Fountoukis et al. (2012), the concentrations of the smallest particles are clearly over-predicted as compared with the semi-empirical schemes. We will summarize this information for all the simulations in two supplementary tables to the revised manuscript. The reason the simulation ACDC-RADM-DE was shown in Fig. 2 of the original manuscript was to facilitate comparison with Fountoukis et al., 2012 with only the NPF mechanism changing. To avoid confusion, we will show ACDC-TUV-DE in Fig. 2 of the revised manuscript (see Fig. R1 below).

It should be noted, however, that the main point of the manuscript is not to prove that the ACDC-based nucleation scheme is more successful than the past semi-empirical schemes in producing the present-day number concentrations (to which the semi-empirical schemes have been fitted) but to rather complement and extend the development of the theoretical understanding of atmospheric particle formation. We probably do not need to resort to semi-empirical schemes for much longer. Having a description that has been evaluated against laboratory data and has e.g. temperature- and RH-dependencies in line with the current theoretical understanding for e.g. extrapolating back to the pre-industrial atmosphere for which we have very little observational data.

|           | Mean                | Mean                | NMB   | NME | Percent     |
|-----------|---------------------|---------------------|-------|-----|-------------|
|           | Observed            | Predicted           | (%)   | (%) | within a    |
|           | (cm -3 ) | (cm -3 ) |       |     | factor of 2 |
|           |                     | Aspvr               | reten |     |             |
| N10       | 2200                | 7420                | 237   | 243 | 33          |
| $N_{50}$  | 1400                | 1270                | -9    | 47  | 65          |
| $N_{100}$ | 580                 | 330                 | -44   | 51  | 57          |
|           |                     | Cabau               | ıw    |     |             |
| N10       | 7700                | 12245               | 59    | 73  | 69          |
| N50       | 4760                | 3300                | -31   | 37  | 81          |
| N100      | 1925                | 1040                | -46   | 50  | 50          |
|           |                     | Hyytia              | ala   |     |             |
| N10       | 2660                | 5570                | 110   | 127 | 48          |
| N50       | 1120                | 1080                | -4    | 61  | 57          |
| N100      | 460                 | 240                 | -48   | 57  | 43          |
|           |                     | Ispi                | ra    |     |             |
| N10       | 7800                | 13240               | 70    | 93  | 62          |
| N50       | 4040                | 3035                | -25   | 41  | 71          |
| N100      | 1725                | 1035                | -40   | 49  | 56          |
|           |                     | Mace Hea            | ad    |     |             |
| N10       | 3200                | 11620               | 263   | 268 | 30          |
| $N_{50}$  | 1825                | 1890                | 4     | 41  | 74          |
| $N_{100}$ | 950                 | 500                 | -49   | 54  | 35          |
|           |                     | Melpi               | itz   |     |             |
| N10       | 9620                | 20700               | 115   | 143 | 46          |
| N50       | 4360                | 3135                | -28   | 37  | 81          |
| $N_{100}$ | 1740                | 780                 | -55   | 56  | 40          |
|           |                     | Vavih               | ill   |     |             |
| N10       | 3580                | 11310               | 216   | 224 | 24          |
| N50       | 1900                | 2010                | 6     | 36  | 84          |
| N100      | 785                 | 550                 | -30   | 41  | 60          |
|           |                     | Overa               | all   |     |             |
| N10       | 5100                | 11540               | 127   | 145 | 44          |
| N50       | 2600                | 2170                | -18   | 41  | 72          |
| N100      | 1100                | 610                 | -45   | 51  | , 2
49   |

Table R1. The statistics of the agreement between the ACDC-RADM-DE and the in-situ observations.

|        |       |             |        | $N_4$      |            |      | N10        |            |      | N160-1040  | )          |
|--------|-------|-------------|--------|------------|------------|------|------------|------------|------|------------|------------|
|        |       |             | R      | NMB
(%) | NME
(%) | R    | NMB
(%) | NME
(%) | R    | NMB
(%) | NME
(%) |
| Estar  | ACDC- | <2km        | 0.18   | 1005       | 1006       | 0.20 | 215        | 224        | 0.40 | -80        | 80         |
| Faicon | DE    | 2-11
km  | 0.49   | 901        | 930        | 0.58 | 249        | 283        | 0.77 | -74        | 81         |
|        | ACDC- | <2km        | 0.08   | 935        | 939        | -    | -          | -          | 0.25 | -68        | 70         |
| ВАе    | DE    | 2-6.4
km | - 0.11 | 420        | 529        | -    | -          | -          | 0.56 | -54        | 78         |

Table R2. The statistics of the agreement between the ACDC-TUV-DE and the aircraft observations.

Figure R1. Scatter plots of predicted vs. measured number concentrations of particles larger than 10 (top), 50 (middle) and 100 (bottom) nm in diameter for ACDC-TUV-DE.

2) Unsubstantiated claims: The paper concludes with, "Overall, we consider our results very promising: a NPF scheme based on first-principles theory and no artificial scaling is shown to be a promising alternative to semi-empirical approaches in the description of particle formation in large scale atmospheric models." There is similar discussion earlier in the conclusions, "reasonable particle concentrations". Obviously, "very promising" and "reasonable" are subjective statements, and different people would view the 10x boundarylayer overprediction for N4 as not "very promising" or "reasonable". We can use Westervelt et al. (2014) for comparison, where they compare simulations using scaled Napari (1E-5 in this case) versus unscaled Napari (also "first principles", and in the unscaled case, without a tuning factor). Table 3 and Figure 2 in Westervelt shows that N10 is only 2x higher in the boundary layer for the unscaled Napari simulation relative to the scaled Napari simulation. . . similar to what was found in the current manuscript in review between ACDC and scaled Napari (1E-6 in the current manuscript). Thus, the overpredictions by the ACDC scheme relative to the scaled Napari in this current manuscript is not unlike comparing scaled and unscaled Napari. I would auess that the nucleation rates predicted by ACDC must be many orders of magnitude faster than scaled Napari in order to get N4 to be 10x too high given all of the microphysical dampening between nucleation rates and N4. While I know that Napari cannot be right for the right reasons, it is very surprising that ACDC does so poorly here considering that it does ok relative to CLOUD measurements (is there a mistake in the lookup tables? are these formation rates per m3 rather than per cm3?). What are the mean nucleation rates predicted at some sites where nucleation rates have been well-characterized (e.g. Hyytiala)?I agree that missing condensible material may be part of the issue here (adding more would likely lower N4 and increase N100). D'Andrea et al. (2013) lowered N10 by about 20-50% over Europe by adding 100 Tg yr-1 globally of non-volatile SOA correlated with anthropogenic CO (see Figure 3). Thus, the N10 bias in the current manuscript may be somewhat fixed by missing condensible material. But regardless, please don't make unsubstantiated claims about the current setup since the ACDC rates seem to be similar to the unscaled Napari scheme that we love to hate.

We agree that the wording in the conclusions was too subjective, and we will modify it to the revised manuscript.

It is important to keep in mind, however, that the implementation of the ACDC-based new particle formation scheme, which included no semi-empirical scaling or fitting, is a significant improvement to earlier new particle formation descriptions and not fundamentally comparable to the "unscaled Napari" scheme. With or without a scaling factor, the Napari scheme is by definition not a first principles NPF scheme: it is a parameterization of data calculated with a classical nucleation theory (CNT) –based approach, which contains several assumptions that are known to be particularly poor for describing the sulfuric acid – ammonia – water system. Some of these shortcomings have been recognized already soon after the publication of the Napari parameterization, and attempts to improve the description by Napari et al. have been ongoing for over a decade (see e.g. Anttila et al. 2005, *Boreal Env. Res.*, 10, 511). Among the most important of additional uncertainties is representing the energetics of the system with bulk thermodynamics e.g. assuming complete proton-transfer which is known not to hold for small clusters and results in drastic errors in the formation free energies and internally inconsistent handling of small stable ammonia-sulfuric acid clusters. These shortcomings cause the parameterization to produce unrealistically high formation rates, which has resulted in the need to scale the rates by five or six orders of magnitude.

It cannot be concluded from the reported  $N_{10}$  or  $N_4$  that the formation rates from the unscaled Napari parameterization would be similar to those of the ACDC-based scheme. Figure R2 below shows formation rates as a function of sulfuric acid concentration, interpolated from the lookup table for conditions

corresponding to those of particle formation experiments in the CLOUD chamber (Almeida et al., *Nature* 2013). The green line corresponds to the experimental conditions of Almeida et al. (2013): T = 278 K, [NH3] = 10 pptv, and RH = 38%. The condensation sink is here set to a representative value of  $10^{-3}$  s-1, corresponding approximately to the sink caused by the chamber walls. However, due to the lack of available quantum chemical data for hydrated clusters at the time of the Almeida et al. (2013) work, the ACDC results presented there correspond to RH = 0%, which have also been added to Fig. R1 for comparison (blue line). Comparison to Fig. 1 in Almeida et al. (2013) shows good agreement with the present lookup table. Increasing the RH to 38% increases the ACDC formation rates. However, there is still around six orders of magnitude difference between the ACDC and unscaled Napari formation rates. It can be seen from Fig. R1 that for sulfuric acid concentrations around  $10^6 - 10^7$  cm-3 the ACDC predictions at 38% range from about  $10^{-3}$  to  $1 \text{ cm}^{-3}$  s-1, which are comparable to atmospheric observations, although in their lower end. The real atmosphere is of course a much more complex system with many more chemical compounds and processes involved in the particle formation, and sinks of clusters that are not yet well understood. Thus, it is quite understandable that the predictive power of any particle formation scheme for a fixed set of compounds is poorer in the atmosphere than in the CLOUD chamber.

Finally, it should be noted that it is not the purpose of this paper to prove that the parameterizations by Napari et al. are wrong (their disagreement with laboratory data has been well known for a long time), but rather make the positive point that we are finally converging towards a molecular-level understanding of the atmospheric particle formation process involving sulfuric acid, water and ammonia, and that this detailed theoretical knowledge can even be incorporated in regional and global atmospheric models with results that are comparable to the present semi-empirical approaches.

Figure R2. Formation rates of particles with a mobility diameter of ca. 1.3 nm as a function of gas phase sulfuric acid concentration, as predicted by the ACDC model at 0% or 38% relative humidity (RH) (blue and green curves), as well as the original "unscaled" parameterization by Napari et al. (red curve).

**Specific comments:**

**L62: Adams and Seinfeld (2002) did not use the Napari nucleation scheme.**

The reviewer is correct. The reference to Adams and Seinfeld will be deleted from the corresponding place in the revised manuscript.

**L145: Why not use ACDC with no ammonia for binary nucleation?**

This is a good suggestion; using the same framework also for the binary NPF would of course be consistent. However, the small sulfuric acid–water clusters of the binary system are so unstable (Henschel et al., *J. Phys. Chem.* 2016) that we would need to include in the ACDC modeling clusters that are way beyond the size of the current simulation system (of the order of at least tens of molecules) to produce binary formation rates over all the conditions of the look-up table, e.g. at higher temperatures relevant to the boundary layer. As there is no quantum chemical data available for hydrated clusters of more than four sulfuric acid molecules, we decided to instead use the Vehkamäki parameterization for the binary pathway. It has been shown to compare reasonably well with experiments and is also applied in many large-scale models. Given that we do not expect the binary formation pathway to be critical during the summer time at the relatively low altitudes sampled by PMCAMx-UF and the available measurements, we have opted to leave this parameterization as is. Further, in so doing, we isolate the effects of the ternary pathway, which are critical for boundary and mixing layer conditions.

L160-165 (as well as throughout): These lines come out of nowhere but seem important. Were natural sources not included in PMCAMx-UF before? I didn't figure this out until later in the paper. While there is a test simulation turning natural emissions on, there is never much discussion about this in the paper. Why isn't having natural emissions on the default setting in this paper (with a sensitivity simulation with them off). I realize you chose this because you didn't have natural emissions before, but shouldn't the simulations with natural emissions be expected to better simulate the size distribution? Please give attention to the text and discussion of natural emissions throughout to ensure that it is clear.

This is the first time that the natural particle emissions (these consist of biogenic, marine and wildfire sources) have been included in PMCAMx-UF in Europe for the period of May 2008. However, our main focus in this study is assessment of the implementation of the ACDC-based NPF-scheme. To do this, we decided to keep the emission the same as in the previous study by Fountoukis et al (2012) which we refer to as "Default Emissions", which did not include natural emissions. Including the natural emissions resulted in better agreement with the observations as compared with the default case, especially for the small sizes. The NMB for the surface level values for ACDC-TUV-NE are 113 % for  $N_4$ , -9% for  $N_{10}$ , -45% for  $N_{100}$ . For ACDC-TUV-DE the corresponding numbers are 126%, -18% and -45%. The trend is similar for NME, being 131% ( $N_4$ ), 41% ( $N_{100}$ ), 52% ( $N_{100}$ ) for ACDC-TUV-NE and 145% ( $N_4$ ), 41% ( $N_{100}$ ), 51% ( $N_{100}$ ) for ACDC-TUV-DE. The corresponding comparison for the vertical profiles is given in table R3. We summarize this information (with quantification of the improvements given in two new supplementary tables) to the revised manuscript and do our best to ensure the clarity in the revised text.

|        |            |             | $N_4$  |            |            | Nio  |            |            | N160-1040 |            |            |
|--------|------------|-------------|--------|------------|------------|------|------------|------------|-----------|------------|------------|
|        |            |             | R      | NMB
(%) | NME
(%) | R    | NMB
(%) | NME
(%) | R         | NMB
(%) | NME
(%) |
|        | ACDC-      | <2km        | 0.18   | 1005       | 1006       | 0.20 | 215        | 224        | 0.40      | -80        | 80         |
|        | DE         | 2-11
km  | 0.49   | 901        | 930        | 0.58 | 249        | 283        | 0.77      | -74        | 81         |
| Falcon | ACDC-      | <2km        | 0.17   | 905        | 906        | 0.21 | 173        | 184        | 0.41      | -77        | 77         |
|        | NE         | 2-11
km  | 0.48   | 748        | 779        | 0.56 | 207        | 243        | 0.84      | -66        | 74         |
|        | ACDC-      | <2km        | 0.08   | 935        | 939        | -    | -          | -          | 0.25      | -68        | 70         |
| BAe    | DE         | 2-6.4
km | - 0.11 | 420        | 529        | -    | -          | -          | 0.56      | -54        | 78         |
|        | ACDC-      | <2km        | 0.16   | 808        | 812        | -    | -          | -          | 0.27      | -62        | 65         |
|        | TUV-
NE | 2-6.4
km | - 0.07 | 306        | 418        | -    | -          | -          | 0.52      | -44        | 78         |

Table R3. The statistics of the agreement between the ACDC-TUV-DE and ACDC-TUV-NE, and the aircraft observations.

L246-247 (and this entire paragraph): Why is TUV \*not\* called for cells with clouds thinner than optical depth 5? Don't you want to consider radiative transfer here? Earlier you discuss wanting to have the effects of simulated aerosols on photolysis rates. . . so wouldn't you want this for clear-sky or thin-cloud conditions? Confusing.

This has been a typo. The TUV module is called for optical depth greater than zero. Optical depths smaller than 5 is considered as clear-sky condition with the default cloud-adjustment scheme (i.e. RADM). The text will be modified in the revised manuscript.

L319: "spatial concentration". Do you mean the relative spatial distribution of concentrations? Since you don't show figures for these other simulations, would it make sense to quantify this in some way, e.g. a correlation coefficient between simulations?

Yes, this referred to the spatial distribution of concentrations simulated by ACDC-TUV-DE as compared with that simulated by Napari-RADM-DE. This will be modified for clarity in the revised manuscript. The correlation coefficients for monthly average concentrations throughout the domain between the different simulation cases and the ACDC-TUV-DE as the base case are generally high, ranging from 0.827 (for  $N_{tot}$  for Napari-TUV-DE vs. the base case) to 0.999 (for  $N_{tot}$  and  $N_{100}$  for ACDC-RADM-DE vs. the base case). It is clear that the simulated

concentrations are very similar, with largest differences observed for the smallest particles if the Napari scheme is used. We will summarize these results in the revised manuscript.

Figures: Why is the default simulation shown in figures sometimes ACDC-TUV-DE (most figures) and sometimes ACDC-RADM-DE (Figure 2)?

This presentation was chosen to directly compare the Fig. 2 with Fig. 3 in Fountoukis et al. (2012) to isolate the effect of the NPF scheme. To avoid confusion, we will show results from ACDC-TUV-DE in Fig. 2 of the revised manuscript.

Figures 3 and 4: Why is the height axis log scale? Do we want to focus mostly on the boundary layer?

The vertical resolution of model is finer in lower levels (see Fig. S1 for the model layer resolution). We have used log-scale to see the variation better.

L377-379: "somewhat closer"? Within 1.5x for scaled Napari vs. outside of 10x for ACDC for most of the boundary layer?! Please be quantitative and avoid subjective judgement.

This is a very good point, the statement was unsubstantiated. We will clarify this in the revised manuscript, basing our statements on the statistics given in the supporting information for the revised manuscript (see Table R1 for an example).

General grammar comment: Adverbs modifying an adjective do not need a hyphen (and should not have a hyphen), e.g. "vertically resolved", "chemically resolved", "newly formed", "hourly averaged". There is no ambiguity in the meaning with or without a hyphen. These should be removed. On the other hand, it is extremely useful to hyphenate joint adjectives. For example, "Aitken-mode particles" should have a hyphen as they are particles in the "Aitken mode". There are not "mode particles" that are "Aitken". Same with "chemical-transport models". They are not "transport models" that are "chemical". How about "large scale atmospheric models"? What is a "scale atmospheric model", and what is so large about it? It is commonplace, unfortunately in my opinion, to omit hyphens when the writer thinks the meaning is unambiguous. However, if a writer never hyphenates joint adjectives, we get into trouble when the writer writes, "slow moving van". Is it a "moving van" that is going slow? Or is it a regular van that is "slow moving"? If a writer establishes that they will hyphenate whenever it is appropriate, we would know that "slow moving van" means a "moving van" that is going slow (else they would have written "slow-moving van").

Thank you for this advice on grammar. We will remove hyphens from adverbs modifying an adjective, and make the usage of e.g. large-scale or Aitken-mode uniform throughout the manuscript. We have also consulted two native English speakers (one with American one with British English as the mother tongue) on the use of hyphenation for joint adjectives.

**Reviewer #2:**

This manuscript presents the implementation of a new ternary H2SO4-H2O-NH3 parameterization, into the PMCAMx-UF model. The authors explore the ability of the model to reproduce observed number concentration during May 2008, when the intensive observation period of EUCAARI project took place. Apart from the testing of the new parameterization, sensitivity tests using the scaled Napari parameterization and sensitivity to the radiation scheme and natural emissions were performed. The topic and overall approach fits with GMD; therefore, I am in favor of accepting this work for publication in GMD after the authors have addressed the issues summarized below.

We thank the reviewer for his /her encouraging comments. Below you can find our point-by-point responses to all the issues raised.

While several sensitivity tests are done, the paper lacks a proper statistics for each test. Figure 2 shows the results coming from ACDC-RADM-DE sensitivity study, however no information is given for the other studies presented in Table 1. It would be nice to see some numbers (r, over/under estimation factor, bias), to endorse the statement "Overall, we consider our results very promising: a NPF scheme based on first-principles theory and no artificial scaling is shown to be a promising alternative to semi-empirical approaches in the description of particle formation in large scale atmospheric models."

This issue was also raised by Prof. Pierce in his review. Table R1 below shows an example statistics for the results presented in Fig. 2 of the original manuscript. We have calculated similar statistics for all simulations and will add them as supplementary information and summarize them in the revised manuscript.

|           | Mean                | Mean                | NMB   | NME | Percent     |
|-----------|---------------------|---------------------|-------|-----|-------------|
|           | Observed            | Predicted           | (%)   | (%) | within a    |
|           | (cm -3 ) | (cm -3 ) |       |     | factor of 2 |
|           |                     | Aspvi               | reten |     |             |
| N10       | 2200                | 7420                | 237   | 243 | 33          |
| $N_{50}$  | 1400                | 1270                | -9    | 47  | 65          |
| $N_{100}$ | 580                 | 330                 | -44   | 51  | 57          |
|           |                     | Cabaı               | ıw    |     |             |
| N10       | 7700                | 12245               | 59    | 73  | 69          |
| N50       | 4760                | 3300                | -31   | 37  | 81          |
| N100      | 1925                | 1040                | -46   | 50  | 50          |
|           | -                   | Hyyti               | ala   |     |             |
| N10       | 2660                | 5570                | 110   | 127 | 48          |
| N50       | 1120                | 1080                | -4    | 61  | 57          |
| N100      | 460                 | 240                 | -48   | 57  | 43          |
|           |                     | Ispi                | ra    |     |             |
| N10       | 7800                | 13240               | 70    | 93  | 62          |
| N50       | 4040                | 3035                | -25   | 41  | 71          |
| N100      | 1725                | 1035                | -40   | 49  | 56          |
|           |                     | Mace Hea            | ad    |     |             |
| N10       | 3200                | 11620               | 263   | 268 | 30          |
| $N_{50}$  | 1825                | 1890                | 4     | 41  | 74          |
| $N_{100}$ | 950                 | 500                 | -49   | 54  | 35          |
|           |                     | Melp                | itz   |     |             |
| N10       | 9620                | 20700               | 115   | 143 | 46          |
| N50       | 4360                | 3135                | -28   | 37  | 81          |
| N100      | 1740                | 780                 | -55   | 56  | 40          |
|           |                     | Vavih               | uill  |     |             |
| N10       | 3580                | 11310               | 216   | 224 | 24          |
| N50       | 1900                | 2010                | 6     | 36  | 84          |
| N100      | 785                 | 550                 | -30   | 41  | 60          |
|           |                     | Overa               | all   |     |             |
| N10       | 5100                | 11540               | 127   | 145 | 44          |
| N50       | 2600                | 2170                | -18   | 41  | 72          |
| N100      | 1100                | 610                 | -45   | 51  | 49          |

Table R1. The statistics of the agreement between the ACDC-RADM-DE and the in-situ observations.

L82: Should be "Matsui et al., 2011, 2013", not "Matsui et al., 2011a, 2013c".

Thank you for pointing this out, we will modify this in a revised version of the manuscript.

L88-90: Matsui et al., 2013 study, already mentioned by the authors, have also assessed the ability of WRF-Chem to reproduce the vertical profile of observed Aitken particles for South Asia.

Thank you for pointing this out, we will modify the manuscript accordingly.

L133-134: Should be "Yu et al., 2006", not "Yu et al., 2006a".

Thank you for pointing this out, we will correct this in the revised manuscript.

L210-213: I assume that if the H2SO4, NH3, RH, temperature and condensation sink are not falling into the mentioned range, the Vehkamaki et al., 2002, parameterization is applied. Is that right?

The Vehkamaki et al. (2002) parameterization is called as long as the  $H_2SO_4$  concentration is greater than  $10^4$  cm-3. As mentioned in section 2.1, the ternary and binary pathways are operating simultaneously. The ternary pathway is called only if the vapor concentrations of both  $H_2SO_4$  and  $NH_3$  are above the lower limit of the lookup table. At lower concentrations the formation rate is practically zero, as the boundaries of the lookup table are chosen so that they should cover the atmospherically relevant range. When the ternary pathway is called, the  $H_2SO_4$ ,  $NH_3$ , RH, temperature and condensation sink are limited to the bounds of the ACDC lookup table if any of the parameters fall outside of the boundaries. Although we did not count the frequency of any exceedances above or below the lookup table bounds, we are confident that these exceedances are few since the bounds are so large compared to atmospherically relevant conditions. Furthermore, even if e.g. the vapor concentrations exceed the upper limits, the rate is likely already converged to a plateau, making it safe to use the values at the limits. We will clarify this in the revised manuscript.

L330-336: The authors show the scatter plots of predicted PNC using ACDC-RADMDE simulation vs observed PNC in several size ranges. Yet, at line 303 they state that ACDC-TUV-DE is the baseline simulation. Do they have any particular reason not to present the results coming from the default simulation? As can be seen in Table 1 the differences between the ACDC-RADM-DE and ACDC-TUV-DE simulations are minors. Furthermore, they use the ACDC-TUV-DE simulation results for the following plots. A little bit confusing.

The reason for presenting ACDC-RADM-DE in Fig. 2 of the original manuscript was to show a direct comparison with Fig. 3 of Foutoukis et al. (2012) with the only difference being the different NPF scheme. To avoid confusion, we will show results from ACDC-TUV-DE in Fig. 2 of the revised manuscript.

L373-379: Could you give an explanation why N4 concentration increases in the upper boundary layer for ACDC-TUV-DE simulation? May you could present the particle formation rates for the ACDC-TUV-DE and Napari-TUV-DE simulation. Also, could you give an overestimation factor?

This is most likely due to the simultaneous drop in temperature and concentrations of larger particles (i.e. the coagulation sink for the newly-formed particles), which enhance both the particle formation rate and their survival. In fact, the temperature dependence is one of the key differences between the new ACDC-based NPF representation and the previous semi-empirical approaches: while the latter do not usually have an explicit temperature dependence of the formation rates, the former does. The ACDC-TUV-DE simulations over-predict  $N_4$  with a factor of 8-10 below 2 km and 4-9 above 2 km (see also responses to Prof. Pierce). We will revise the manuscript accordingly to clarify these issues.

L384-389: An index of agreement will sustain "the scaled Napari NPF scheme agrees reasonably well with the observations throughout the atmospheric column" and "reasonably well" statements.

We agree that these are subjective statements that need to be backed up by quantitative data. As discussed above, we will add two supplementary tables (showing the NMB, NME and correlation coefficients for all the simulations vs. in-situ and aircraft data, respectively) and summarize these statistics in the text. We have also calculated indexes of agreement for all our data (surface level and aircraft). For the Napari-TUV-DE simulations these values range from about 0.2 to 0.7 for all the size ranges throughout the atmospheric column, while for the ACDC-TUV-DE shows a large difference between the in-situ (0.3-0.6 for all sizes) and aircraft (0.02-0.7 for all sizes, with the poorest agreement for the smallest particles) data sets. We hope the inclusion of more statistical metrics will now do the job and will also go through the revised manuscript to remove all unnecessary subjective statements.

L433\_435: The following sentence for a more scientifically sound expression should be rephrased: "We believe this is the first time that reasonable particle concentrations have been produced in a large-scale atmospheric a

Good point. We will modify the revised manuscript accordingly.

The authors should be more restrictive in using "reasonably well", "are somewhat overpredicted by the ACDCbased NPF scheme", "very promising" statements due to the fact that the lack of statistics throughout the paper does not sustain their claims.

Besides adding the aforementioned more quantitative analysis to support these statements, we will limit the usage of the phrases in question in the revised manuscript.

**Implementation of state-of-the-art ternary new particle formation scheme to the regional chemical transport model PMCAMx-UF in Europe**

E. Baranizadeh1, B. N. Murphy2,3, J. Julin1,2, S. Falahat2,4, C. L. Reddington5, A. Arola6, L. Ahlm2, S. Mikkonen1,
C. Fountoukis7, D. Patoulias8, A. Minikin9, T. Hamburger10, A. Laaksonen1,11, S. N. Pandis8,12,13, H. Vehkamäki14,
K. E. J. Lehtinen1,6, I. Riipinen2

1Department of Applied Physics, University of Eastern Finland, Kuopio, Finland 2Department of Environmental Science and Analytical Chemistry (ACES), Stockholm University, Stockholm, Sweden 3Now at the National Exposure Research Laboratory, US Environmental Protection Agency, Research Triangle Park, USA 4Now at the Swedish Meteorological and Hydrological institute (SMHI), Norrköping, Sweden 5Institute for Climate and Atmospheric Science, School of Earth and Environment, University of Leeds, Leeds, UK 6Finnish Meteorological Institute, Kuopio, Finland 7Qatar Environment and Energy Research Institute (QEERI), Hamad Bin Khalifa University (HBKU), Qatar Foundation, Doha, Qatar 8Department of Chemical Engineering, University of Patras, Patras, Greece 9German Aerospace Agency DLR, Oberpfaffenhofen, Germany 10Norwegian Institute for Air Research (NILU), Oslo, Norway 11Climate research Unit, Finnish Meteorological Institute, Helsinki, Finland 12Institute 
[revised manuscript text omitted]